# Proteomic Analysis of Human iPSC-Derived Neural Stem Cells and Motor Neurons Identifies Proteasome Structural Alterations

**DOI:** 10.3390/cells12242800

**Published:** 2023-12-08

**Authors:** Iñaki Álvarez, Adrián Tirado-Herranz, Belén Alvarez-Palomo, Jordi Requena Osete, Michael J. Edel

**Affiliations:** 1Departament de Biologia Cellular, Institut de Biotecnologia i Biomedicina, Universitat Autònoma de Barcelona, Fisiologia i Immunologia, 08193 Barcelona, Spain; inaki.alvarez@uab.cat (I.Á.); igalvarez3@gmail.com (A.T.-H.); 2Banc de Sang i Teixits, Edifici Dr. Frederic Duran i Jordà, Passeig Taulat, 116, 08005 Barcelona, Spain; abalvarez@bst.cat; 3Department of Medical Genetics, Oslo University Hospital, 0450 Oslo, Norway; 4NORMENT, Institute of Clinical Medicine, University of Oslo, 0316 Oslo, Norway; 5Division of Mental Health and Addiction, Oslo University Hospital, 4956 Oslo, Norway; 6Department of Anatomy and Embryology, Faculty of Medicine, Universitat Autònoma de Barcelona, 08193 Barcelona, Spain; 7Discipline of Medical Sciences and Genetics, School of Biomedical Sciences, University of Western Australia, Perth 6009, Australia

**Keywords:** proteomics, induced pluripotent stem cells, differentiation, neural stem cells, motor neurons, 26S proteasome, proteasome

## Abstract

Background: Proteins targeted by the ubiquitin proteasome system (UPS) are identified for degradation by the proteasome, which has been implicated in the development of neurodegenerative diseases. Major histocompatibility complex (MHC) molecules present peptides broken down by the proteasome and are involved in neuronal plasticity, regulating the synapse number and axon regeneration in the central or peripheral nervous system during development and in brain diseases. The mechanisms governing these effects are mostly unknown, but evidence from different compartments of the cerebral cortex indicates the presence of immune-like MHC receptors in the central nervous system. Methods: We used human induced pluripotent stem cells (iPSCs) differentiated into neural stem cells and then into motor neurons as a developmental model to better understand the structure of the proteasome in developing motor neurons. We performed a proteomic analysis of starting human skin fibroblasts, their matching iPSCs, differentiated neural stem cells and motor neurons that highlighted significant differences in the constitutive proteasome and immunoproteasome subunits during development toward motor neurons from iPSCs. Results: The proteomic analysis showed that the catalytic proteasome subunits expressed in fibroblasts differed from those in the neural stem cells and motor neurons. Western blot analysis confirmed the proteomic data, particularly the decreased expression of the β5i (PSMB8) subunit immunoproteasome in MNs compared to HFFs and increased β5 (PSMB5) in MNs compared to HFFs. Conclusion: The constitutive proteasome subunits are upregulated in iPSCs and NSCs from HFFs. Immunoproteasome subunit β5i expression is higher in MNs than NSCs; however, overall, there is more of a constitutive proteasome structure in MNs when comparing HFFs to MNs. The proteasome composition may have implications for motor neuron development and neurodevelopmental diseases that warrant further investigation.

## 1. Introduction

The discovery of a method to make human induced pluripotent stem cells (iPSC) by Yamanaka in 2007 created the opportunity to model human disease and development at a new level [1]. However, successful and accurate disease modeling, as well as the application of human iPSCs for human disease, is hampered by genetic quality. There have been many improvements to the original cell reprogramming protocol, including the use of non-viral methods, such as synthetic mRNA transfection, for the delivery of reprogramming factors to cells [2]. Also, replacing the c-Myc reprogramming factor with Cyclin D1, which repairs DNA breaks that occur during the reprogramming process, significantly improves the genetical stability of iPSCs and neural stem cells (NSCs) [3]. The use of these advanced iPSCs and NSCs results in more accurate conclusions when applying them as a human disease model in vitro [3].

The peptide products of proteasome digestion are the source of peptides presented by major histocompatibility class I molecules (MHC-Is) on cells that can be detected by CD8 T lymphocytes. Thus, the expression of different types of proteasomes in different cells will likely change the immunopeptidomes bound to MHC-Is on the cell surface. MHC-Is, which present peptides broken down by the proteosome, have been implicated in neuronal plasticity, regulating synaptic density and axonal regeneration in the central and peripheral nervous system during development and in brain diseases [4,5]. The mechanisms governing these effects are largely unknown.

The most abundant proteasome in the cell is the 26S proteasome, which is composed of a catalytic core and the 20S proteasome bound at the ends to a regulatory complex, which is called the 19S proteasome. Proteins targeted by the ubiquitin proteasome system (UPS) are identified for degradation by the proteasome. The 20S proteasome has a barrel shape with seven stacking heptameric rings. The outer rings are structural and are composed of seven α subunits. The two inner rings are composed of β subunits; three of them have catalytic functions: β1, β2 and β5. Most cells principally express this constitutive or standard proteasome (CP). In addition, other regulatory complexes can replace the 19S complex. In the presence of IFN-γ, these subunits can be replaced by others called β1i, β2i and β5i, forming the immunoproteasome (IP) [6]. The assembly of this structure is highly regulated [7]. Therefore, the presence of an IP structure in a cell indicates that the cell is experiencing an inflammatory response.

Dysfunction of the UPS system has been implicated in neurodegenerative diseases, such as amyotrophic lateral sclerosis (ALS) in motor neurons (MNs) [8,9,10,11,12,13]. Mutations in the key UPS regulator UBA1 can cause the juvenile motor neuron disease spinal muscular atrophy (SMA) [14,15]. In addition, spinal MNs appear to present a higher susceptibility than other cell types to UPS stress [16].

The CP is the most abundant proteasome present in most body tissues, except in lymphoid tissues, where the IP is expressed in a higher amount [17]. In the nervous system, the IP is practically absent in the brain and is more abundant in nerves, but in both, the constitutive proteasome is the most abundant proteasome type [17].

Here, we describe the use of human iPSCs differentiated into neural stem cells and then motor neurons as a developmental model to better understand the structure of the proteasome in developing motor neurons.

## 2. Materials and Methods

### 2.1. Human iPSC Differentiation to Motor Neurons

Human iPSC clones were previously generated either by synthetic mRNA transfection methods or retroviral methods and were used for this project [3]. Briefly, for synthetic mRNA-made iPSCs, human foreskin fibroblasts (Millipore HFF) were reprogrammed to induced pluripotent stem cells (iPSCs) with messenger mRNA transfections and characterized for pluripotency markers [3]. Motor neurons were differentiated from iPSC-derived neural stem cells using the motor neuron induction medium, as described in Figure 1. Briefly, iPSCs were brought into culture 1 week before (day 7) and split into 6-well plates to start the differentiation process described in Figure 1. Human foreskin fibroblasts, synthetic-mRNA-derived iPSCs and neural stem cells made from those iPSC were analyzed by global proteomics. The cell lines, SP15 and CHB fibroblasts, iPSC1 and iPSC2 (retroviral-derived iPSCs from SP15 and CHB fibroblasts) were added for further validation studies, and MNs from synthetic-mRNA-derived iPSCs were included for Western blot validation experiments of identified proteins from the global proteomics analysis (Table 1).

### 2.2. Proteomics

#### 2.2.1. Proteomics Sample Preparation and Measurement

The protein concentration of each sample was determined by BCA assay (Pierce). A total of 10 µg of each sample was digested by trypsin and Lys-C endopeptidases by applying a modified filter-aided sample preparation protocol as described previously [18]. Briefly, protein lysates were denatured by adding 8 M urea buffer at pH 8.5 prior to reduction with dithiothreitol and carbamidomethylation with iodoacetamide. Unfolded proteins were centrifuged on 30 kDa centrifugal filters using a Vivacon 500 device (Sartorius, Göttingen, Germany). After washing with 8M urea buffer and 50 mM ammonium bicarbonate buffer, the proteins on the filters were digested with 0.5 µg of Lysyl Endopeptidase (Wako, Osaka, Japan) for 2 h followed by tryptic digestion (1 µg of trypsin, Promega, Madison, WI, USA) overnight at 37 °C. Peptides were collected by centrifugation through the filter, acidified by trifluoroacetic acid and stored at −20 °C. The samples were measured on a Q-Exactive HF mass spectrometer online coupled with an Ultimate 3000 nano-RSLC device (Thermo Scientific, Waltham, MA, USA) in the data-independent acquisition (DIA) mode as described previously [19]. Briefly, the samples were loaded onto a C18 trap column (300 μm inner diameter × 5 mm, packed with AcclaimPepMap100 C18, 5 μm, 100 Å; LC Packings, Sunnyvale, CA, USA) prior to separation of the peptides in a 105 min non-linear acetonitrile gradient (from 7 to 41% ACN in 0.1% formic acid) on an analytical column (AcquityUPLC M-Class HSS T3 Column, 1.8 μm, 75 μm × 250 mm; Waters, Milford, MA, USA). The data-independent acquisition method contained an MS full-scan at a 120,000 resolution from 300 to 1650 m/z with an AGC target of 3 × 10^6^ and a maximum injection time of 120 ms, which was followed by 37 MSMS fragment scans in DIA windows of varying sizes covering the whole m/z range at a resolution of 30,000 applying a normalized collision energy of 28.

#### 2.2.2. Quantitative MS Analysis

The DIA LC–MS/MS data set was analyzed using a Spectronaut Pulsar device (Biognosys, Schlieren, Switzerland) as described previously [19]. The Spectronaut HTRMS converter software v16 was used to convert the raw files. The default settings in the software were applied with the following additions: Searches were performed using a human spectral meta-library, which was generated by analyzing data-dependent acquisition runs from a variety of human samples in the Proteomics Discoverer software (Version 2.1, Thermo Scientific). Proteotypic peptide and protein identifications were filtered for a false discovery rate of <1%. Matching between runs was enabled with the data filtering function set to *q*-value percentile mode applying the 20% setting. The resulting peptide and protein quantifications in the individual samples were exported and used for calculations of fold-changes and statistical values.

### 2.3. Immunofluorescence

Motor neurons grown in chamber slides (ThermoFisher, #177437) were fixed with 4% paraformaldehyde in PBS for 20 min, permeabilized with 0.2% Triton X-100 in PBS and blocked in 6% donkey serum for 1 h. Then, the cells were incubated with primary and secondary antibodies. The primary antibodies and dilutions used were as follows: anti-Tuj1 (Biolegend, San Diego, CA, USA, MMS-435P-100, 1:500), anti-Olig2 (R&D systems, Minneapolis, MN, USA, AF2418), anti-Nestin (Biolegend, 841801, 1:200) and anti-Sox2 (CalBiochem, San Diego, CA, USA, sc1002, 1:100). The secondary antibodies used were all from the Alexa Fluor Series from Invitrogen (dilution 1:200). Nuclei were stained with 4′,6-diamidino-2-phenylindole (DAPI, Roche, Basel, Switzerland, 10236276001), and then cover glasses (VWR, ECN631-1575) were mounted on top with Fluoromount-G (Invitrogen, Waltham, MA, USA, #00-4958-02). Images were taken using a Leica SP5 (Mannheim, Germany) confocal microscope (Mannheim, Germany) and were processed using the Fiji software 2.9.0 version.

### 2.4. Real Time PCR

Total mRNA was isolated using an Ambion RNA purification columns kit (#12183018), and 500 ng was used to synthesize cDNA using a SensiFAST cDNA synthesis kit (Bioline, Essex, UK, BIO65053). One microliter of the reaction was used to quantify gene expression by quantitative PCR as previously described [20]. The primer sequences used were as follows: Hgapdh Fw: 5′-GCACCGTCAAGGCTGAGAAC-3′, Hgapdh Rv: 5′-AGGGATCTCGCTCCTGGAA-3′; hChat Fw: 5′-AACGAGGACGAGCGTTTG-3′, hChat Rv: 5′-TCAATCATGTCCAGCGAGTC-3′; hHoxB4 Fw: 5′-GTCGTCTACCCCTGGATGC-3′, hHoxB4 Rv: 5′-TTCCTTCTCCAGCTCCAAGA-3′; hNkx6.1 Fw: 5′-ATTCGTTGGGGATGACAGAG-3′, hNkx6.1 Rv: 5′-CCGAGTCCTGCTTCTTCTTG-3′ and hPeripherin Fw: 5′-AGACCATTGAGACCCGGAAT-3′, hPeripherin Rv: 5′-GGCCTAGGGCAGAGTCAAG-3′. For proteasomes, they were as follows: β1i: Fw: accaaccggggacttacc, Rv: tcaaacactcggttcaccac; β2i: Fw: ggttccagccgaacatga, Rv: gcccaggtcacccaagat and β5i: Fw: accccgcgtgacactact, Rv: gggactggaagaattctgtgg. Relative quantification was determined according to the ΔΔCT method. Statistical analysis was performed using Student’s T test on between three and six clones.

### 2.5. Karyotype Analysis

To stain chromosome G bands, cells fixed with methanol: acetic acid (3:1) were dyed with Wright: Sorensen buffer (1:3). Twenty metaphases were assessed per sample, and chromosomes were classified using the Ikaros software G4.

### 2.6. Electrophysiology

For whole-cell patch-clamp experiments, iPSC-derived motor neurons grown in round cover glasses were kept at room temperature in HEPES-based ACSF composed of 135 Mm NaCl, 2 Mm KCl, 2 Mm CaCl_2_, 1 Mm MgSO_4_, 10 Mm HEPES and 10 Mm D-glucose at pH 7.35 and with 300–310 mOsm/L. Sodium and potassium currents were measured using whole-cell patch-clamp electrophysiology recordings. In several cells, sodium currents were inhibited by tetrodotoxin (TTX). MN firing action potentials were also recorded.

### 2.7. Western Blot

Cell pellets were resuspended in 200 μL of lysis buffer (LB: 50 Mm Tris-HCl pH 7.4, 150 Mm NaCl, 1% Triton X-100, Protease Inhibitor cocktail (complete^TM^ Tablets Mini, Roche)) and homogenized using 30 g syringe needles, and they were then incubated for 1h on ice. All the samples were then centrifuged (13,000× *g*, 10 min, 4 °C), and the supernatants were taken, quantified and stored at −20 °C. The protein concentration of the cell lysates was calculated using a BCA quantification kit (Pierce™ BCA Protein Assay Kit (Thermo Scientific™)) in a colorimetric plate reader (Victor3™ Plate Reader (PerkinElmer ™, Waltham, MA, USA)). About 10 ug of the protein samples was separated by electrophoresis under denaturing conditions (SDS-PAGE) in a 14% polyacrylamide gels. Proteins were transferred to polyvinylidene membranes (Immun-Blot^®^ PVDF Membrane (BioRad™, Hercules, CA, USA)) that were previously activated for 5 min with methanol for a maximum of 45 min at 100 V. The membranes were subsequently incubated in blocking solution (BS: PBS, 0.1% Tween 20, 5% skimmed milk powder) for 1h with gentle shaking. Then, the membranes were washed 3 times with T-PBS (PBS, 0.1% Tween 20) for 5 min with shaking and then incubated with the primary antibody at a 1/1000 dilution in T-PBS. The membranes were incubated overnight under shaking conditions at 4 °C. After 3 washes with T-PBS, the secondary antibodies were added at a concentration of 1/10,000 diluted in T-PBS, and the membranes were incubated for 1h under shaking conditions at room temperature. Finally, the membranes were washed 4 times with T-PBS for 5 min under shaking conditions. Finally, a detection reagent was applied (1:1 Clarity Western ECL kit Blotting Substrate (BioRad™)), and proteins were detected by chemiluminescence using a ^®^ VersaDoc ™ Molecular Imager (BioRad™) and the Quantity One^®^ 1-D analysis software. The Fiji software (ImageJ, version 1.53t) was used for the densitometry analysis.

The antibodies used for the Western blot analysis were as follows: anti-β1 (PSMB9) goat monoclonal IgG2a antibody (Santa Cruz Biotechnology^TM^, Santa Cruz, CA, USA); anti-β1i (PSMB6) polyclonal rabbit IgG antibody (Santa Cruz Biotechnology^TM^); anti-β2 (PSMB7) mouse monoclonal IgG1 antibody (Santa Cruz Biotechnology^TM^); anti-β2i (PSMB10) mouse monoclonal IgG2b antibody (Santa Cruz Biotechnology^TM^); anti-β5 (PSMB5) mouse monoclonal IgG2a antibody (Santa Cruz Biotechnology^TM^); anti-β5i (PSMB8) mouse monoclonal IgG1 antibody (Santa Cruz Biotechnology^TM^); MCP21 mouse polyclonal antibody (anti-α2 subunit); αVinculin (MA5-11690) mouse monoclonal IgG antibody (Invitrogen^TM^); ECL αMouse IgG, horseradish-peroxidase-linked whole antibody from sheep (GE Healthcare^TM^, Chicago, IL, USA); Precision ProteinTM StrepTactin-HRP Conjugate 5000x (BioRad^TM^); ECL αRabbit IgG, horseradish-peroxidase-linked whole antibody from donkey (GE Healthcare^TM^) and αSheep/Goat-Immunoglobulin peroxidase (AP360) (The Binding Site LTD^TM^, Birmingham, UK).

## 3. Results

### 3.1. Human iPSC Differentiation to Motor Neurons

The synthetic-mRNA-generated iPSC clones were successfully differentiated into motor neurons (MN) within 50 days with a defined protocol with a good morphology based on phase contrast photos (Figure 1). The karyotype of the MNs revealed no major genetic integrity chromosome aberrations (Figure 1C). RT-PCR characterization captured the upregulation of specific MN maturation markers such as Chat, HoxB4, Nkx6.1 and Peripherin at different time points of differentiation (Figure 1D), and immunofluorescence confirmed a positive protein expression of TUJ1, NESTIN, SOX2 and OLIG2 (Figure 2), indicative of the successful MN maturation achieved. No significant differences were found between the lines. Next, we assessed the functional activity of the MNs by patch-clamp electrophysiological recordings (Figure 3). The neurons demonstrated the generation of action potentials (APs) upon electrical stimulation (Figure 3A,B).

### 3.2. Results of Proteomic Analysis of HFFs, iPSCs and NSCs

#### 3.2.1. Analysis of HFFs vs. iPSCs

A total of 2192 proteins were identified, and 1600 proteins were found to be differentially regulated upon the formation of iPSCs from HFFs. A total of 600 proteins were downregulated (Appendix A; Figure 4 (left)) and were mainly involved in biological processes such as cell proliferation, RNA splicing, protein synthesis, cell signaling, sumo, ubiquitination (ligases, de-ubi), HAT1, HDAC, methyltransferases, HSP, Hif1a, metabolic changes RNA-polymerase-II-related processes (Appendix A: Excel table of proteomics).

Interestingly, a variation in several UPS proteins was observed. First, we found a downregulation of PSME1 (PA28alpha) and PSME2 (PA28beta) upon iPSC differentiation. These are IFNγ-induced proteins that are composed of a regulatory complex (PA28) that can replace the 19S regulatory complex at the ends of the 20S proteasome to form the immunoproteasome (IP). In contrast, a 2.4-fold upregulation of PSME3 protein (PA28-gamma) and a 1.8-fold increase in PSMG1 protein were observed in the iPSCs. PSME3 protein has been implicated in cancer by inhibiting c-Myc degradation and is a target gene of NF-κB during bacterial infections. PSMG1 protein enables molecular adaptor activity in the chaperone-mediated protein complex assembly located in the golgi apparatus, endoplasmic reticulum and nucleoplasm. Although fibroblasts mainly express CP, the proteomics analysis showed that immunosubunits (β1i (PSMB9), β2i (PSMB10) and β5i (PSMB8)) were detected in lower amounts in the iPSCs in comparison with the fibroblasts. The opposite was observed for CP subunits (β1 (PSMB6), β2 (PSMB7) and β5 (PSMB5)) (Figure 5). Thus, IFNγ-induced catalytic or regulatory subunits were downregulated in the iPSCs regarding fibroblasts. The downregulation of these proteins, which are antigen-processing-related factors, may affect antigen processing, which can produce a change in the HLA-I peptide repertoire presented to CD8 T lymphocytes.

#### 3.2.2. Analysis of HFFs Compared to iPSCs and NSCs

A total of 800 proteins were found, and a total of 314 proteins were downregulated (Appendix A. Excel file). We found a 1.8-fold upregulation of PSMF1 and a downregulation of PSMD1, PSMD6 and PSMC6 proteins in the NSCs compared to the HFFs (Figure 4 (right)). Although it was not statistically significant, a trend in which IP subunits were increased and CP was decreased in the NSCs in comparison with the iPSCs was observed. In addition, PSMF1, a subunit of the proteasome inhibitor PI31, was statistically increased (Figure 6). This protein inhibits the activation of the 20S proteasome by PA28.

### 3.3. Validation of Constitutive and Immunoproteasome Findings

First, we validated the proteomics data by Western blotting, confirming that the CP subunits (β1 (PSMB6), β2 (PSMB7) and β5 (PSMB5)) were upregulated in the iPSCs compared to the starting HFFs (Appendix A).

To evaluate the expression levels of CP or IP subunits in the fibroblasts, neural stem cells (NSCs) and motor neurons (MNs), Western blot experiments were performed. As expected, the abundance of the CP subunits was high in all cell types, being more expressed in the NSCs and MNs than in the fibroblasts, demonstrating that CP subunit (β1 (PSMB6), β2 (PSMB7) and β5 (PSMB5)) expression is higher in iPSCs, NSCs and MNs than in HFFs (Appendix A and Figure 6). On the other hand, IP subunits were present in a lower amount in all the cell types. In fact, β1i and β2i could not be detected in the fibroblast cells, NSCs or MNs, and they were only detected in the DKB cells, a lymphoblastoid cell line used as control of IP expression (Appendix A). β1i and β2i gene expression was detected by RT-PCR, and the data demonstrated a decrease in both β1i and β2i subunits with the generation of iPSCs or NSCs from HFFs and an increase with differentiation to MNs (Appendix A). The β5i subunit was detected in the fibroblasts, NSCs and MNs in comparison with the DKB cells (Figure 6). The expression level of the β5i subunit was higher in the fibroblasts than in the NSCs or MNs. β5i gene expression was detected by RT-PCR, and the data demonstrated a decrease with the generation of iPSCs or NSCs from HFFs and an increase with differentiation to MNs (Figure 6). We observed a trend in which the level of IP subunits was decreased in the MNs compared to HFFs, while the level of CP subunits increased, suggesting a proteasomal structural change in the MN cells (Figure 6 and Appendix A). Thus, the data indicate that the reprogramming of fibroblasts to iPSCs reduces the abundance of IP and increases the abundance CP, which is progressively recovered during differentiation to MNs.

## 4. Discussion

### 4.1. Proteomics and Western Blot Data Validation in iPSCs and MNs

The Western blot data validated the proteomics analysis and consistently demonstrated that CP subunit (β1 (PSMB6), β2 (PSMB7) and β5 (PSMB5)) expression was higher in the iPSCs, NSCs and MNs than in the HFFs (Appendix A and Figure 6). The data agree with previous results that showed a reduction in IP subunits in iPSCs [21]. On the other hand, IP subunits were present in a lower amount in all the cell types (Figure 6). In fact, β1i and β2i protein could not be detected in the fibroblast cells, NSCs or MNs, and they were only detected in the DKB cells (Appendix A). The β5i subunit was detected in the fibroblasts, NSCs and MNs in comparison with the DKB cells. The expression level of the β5i subunit was higher in the fibroblasts than in the NSCs or MNs (Figure 5). We observed a trend in which the level of IP subunits was increased in the MNs compared to the NSCs, while the level of CP subunits was decreased, suggesting a proteasomal structural change in the MN cells (Figure 6). Therefore, our data suggest that the catalytic activity and specificity of the proteasomes are associated with MN development. Alterations in proteasome subunits expressed in fibroblasts from NSCs and MNs could indicate that the HLA-I immunopeptidomes presented on the cell surface will likely differ, as CP and IP present different protein cleavage specificities.

### 4.2. The Role of the UPS and MHC

The important role of the UPS in pluripotent stem cell survival and motor neuron differentiation has been reported previously [21,22]. This work demonstrated that iPSCs are very sensitive to the proteasome inhibitor MG132, and MNs are more resilient that iPSCs but more sensitive than fibroblasts. In addition, different mutations in the UPS are related with the development of neurodegenerative diseases, further highlighting their importance in neuron survival and health [9,10,11,12,13,14,15].

### 4.3. MHC-I Expression in Disease

One of the roles of the UPS is to produce small peptides to be presented by MHC-Is. Our data suggest that this might be different between patients and transplanted MNs because of the structural change in the proteasome. Recent evidence from independent laboratories has reported the expression of immune-like MHC-I receptors in the central nervous system [4,5]. This work provides some insight into the mechanism of the role of major histocompatibility complex class I proteins in brain development and plasticity [4,5]. The expression of MHC-I molecules and the immunoproteasome is highly increased in the spinal motor neurons of transgenic mice carrying the mutant SOS_1G93A_ during the progression of the disease [23]. Thus, in future studies, it may be relevant to evaluate the role of the immunoproteasome in neurologic disorders using iPSC-derived MNs.

### 4.4. MHC Peptides Technique

The MHC-presented small-peptide repertoire, also referred as immunopeptidome, cannot be determined from the mRNA or protein abundance. The technique used to characterize the immunopeptidome uses mass spectrometry analyses of peptides eluted from MHC complex isolation. However, mass spectrometry is not always sufficient to define the full repertoire of small peptides loaded onto MHC molecules. Moreover, the proinflammatory environment of the spinal cord and the dysregulated protein metabolism of motor neurons in ALS may promote the activation of the IP and the membrane presentation of antigenic peptides recognized as non-self by CD8+ T cells, which then activates a cytotoxic autoimmune response [22,23,24,25]. Taken together, the data confirm that future analysis of the MHC-I immunopeptidomes from neurological cells in different cellular states and in diseased tissues is of great interest. This would help to reveal the role of the proteasome structure during motor neuron development or disease that could allow the discovery of putative targets of the T cell immune response for future treatment strategies.

### 4.5. Misfolded Proteins in Neuronal Disease

The role of misfolded proteins in motor neurons is one of the main neuropathological hallmarks of ALS and is regarded as the primary cause of motor neuron degeneration. Misfolded proteins that cannot be degraded by the constitutive proteasome are directed toward the INF-γ-activated IP [23]. Misfolded proteins in the motor neurons may trigger inflammation through the release of danger-associated molecular pattern molecules (DAMPs), including ROS, HMGB1 and HSPs, which activate glial and immune cells to produce inflammatory cytokines including IFN-γ [22,24]. IFN-γ may then induce the upregulation of the IP in the motor neurons.

## 5. Conclusions

The data revealed an increase in CP subunits and a decrease in IP-associated proteins in MNs compared to HFFs and cells derived from iPSC, suggesting that the HLA-I immunopeptidomes presented on the cell surface will differ, which could have implications in disease modelling and the transplantation of iPSC-derived MNs in the future. Further work is warranted to describe the different small-peptide repertoire presented by MHC-Is in healthy and diseased MNs derived from iPSCs.

## Figures and Tables

**Figure 1 cells-12-02800-f001:**
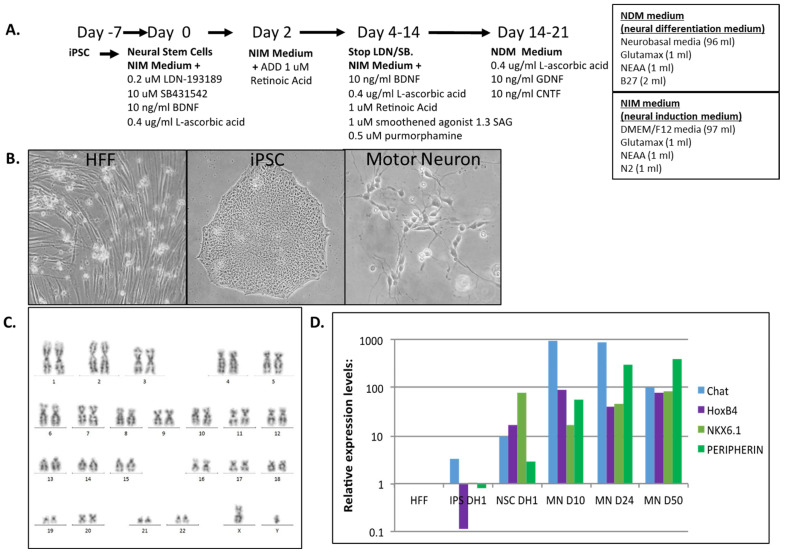
Protocol to generate motor neurons from human iPSCs and their characterization. iPSCs were cultured for 7 days and then differentiated to neural stem cells (NSCs) and motor neurons (MNs). MNs were differentiated up to day 50. (**A**) Differentiation protocol and medias used. (**B**) Phase contrast photographs of fibroblasts, iPSCs, and MNs. (**C**) Karyoptype of MNs. (**D**) RT-PCR analysis of mRNA expression of markers of MNs. HFFs: human foreskin fibroblasts; iPSCs: induced pluripotent stem cells; MN: motor neuron.

**Figure 2 cells-12-02800-f002:**
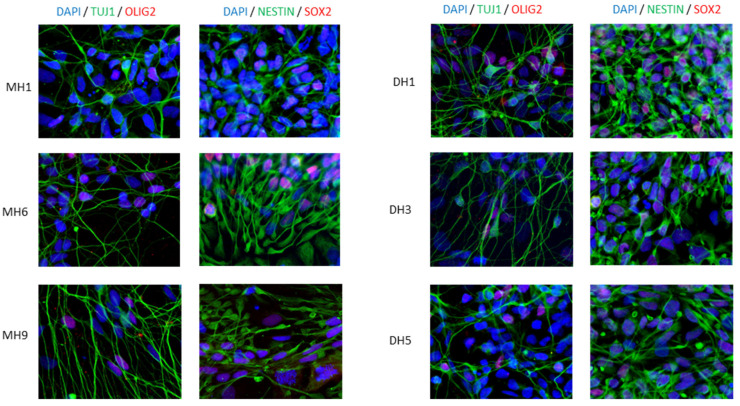
Characterization by immuno-staining protein markers of motor neurons demonstrated high efficiency of differentiation. Each clone, listed in Table 1, was from six previously published human iPSC lines [3]. Mag ×630.

**Figure 3 cells-12-02800-f003:**
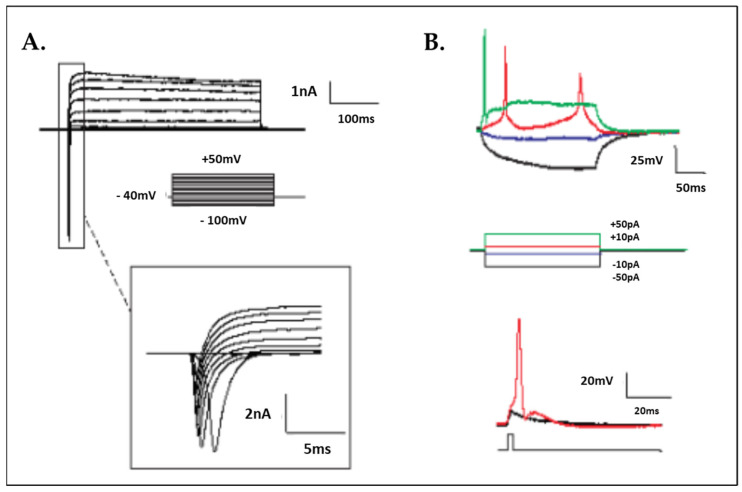
Electrophysiology characterization by patch-clamp analysis demonstrated functional motor neurons. (**A**) Representative traces of voltage-activated Na^+^ and K^+^ currents in iPSC-derived neurons recorded with the whole-cell configuration of the patch-clamp technique. Cells were held at −60 mV, and voltage pulses were applied from −100 to +50 mV to activate the currents. Inset: magnification of the voltage-activated Na^+^ currents recorded. (**B**) Application of 20 nM tetrodotoxin (TTX) blocked inward Na^+^ currents in the recording but did not affect outward K^+^ currents. Traces in black and blue represent control conditions (with/without synaptic blockers), trace in red is 20 nM TTX and trace in green is after the washout of TXX. Insert shows no effect of the outward K^+^ currents. MH1-MH9 and DH1-DH5 are motor neurons from six iPSC clones as listed in Table 1.

**Figure 4 cells-12-02800-f004:**
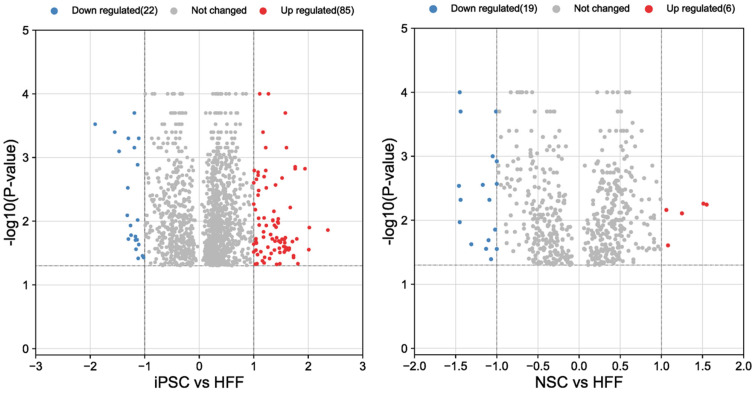
Proteomic analysis: Volcano plots for significant (*p* < 0.05) proteins up- or downregulated between iPSCs/HFFs and NSCs/HFFs. HFFs: human foreskin fibroblasts; iPSCs: induced pluripotent stem cells; NSCs: neural stem cells.

**Figure 5 cells-12-02800-f005:**
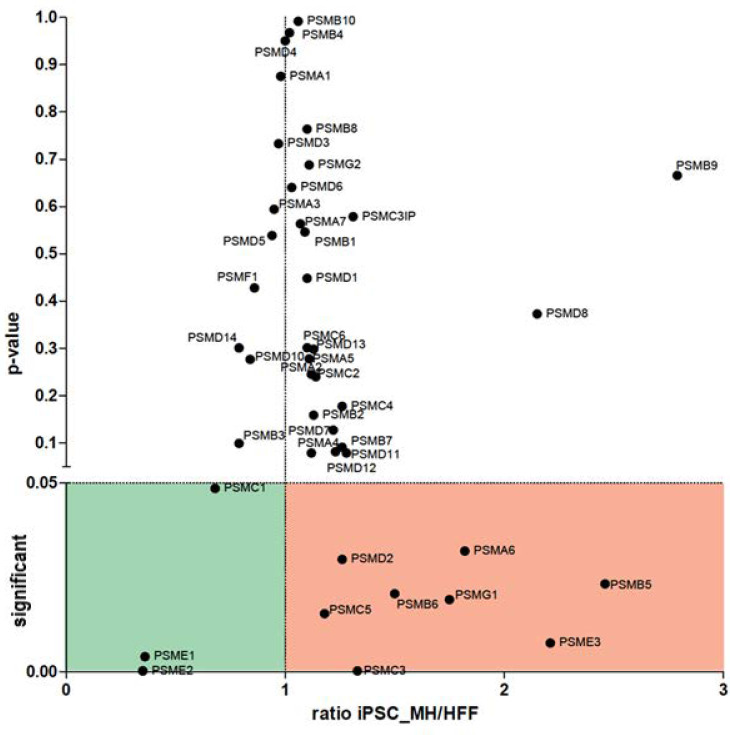
Proteomic analysis: Volcano plot analysis of proteasomal proteins identified in a proteomic screen of parental human foreskin fibroblasts (HFFs) compared to iPSCs, which was derived from these cells taking into account only significantly regulated proteins (*p* < 0.05) identified with at least two peptides.

**Figure 6 cells-12-02800-f006:**
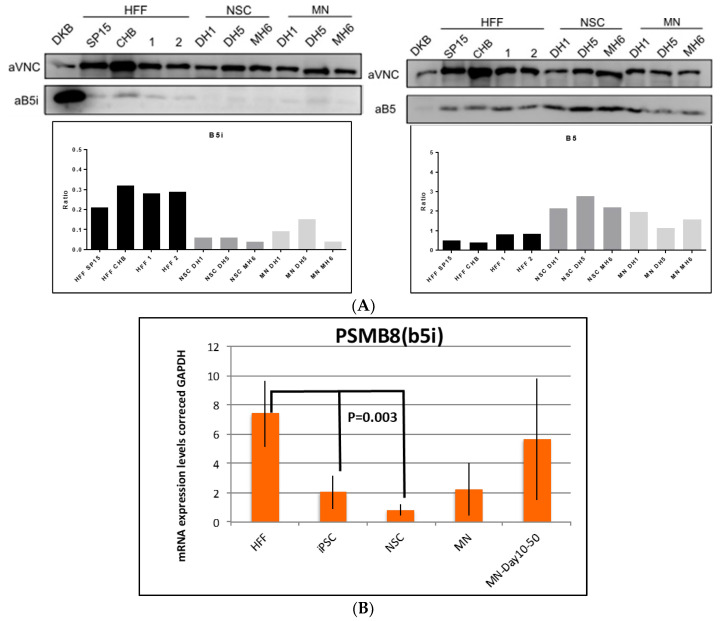
(**A**). Western blot validation for β5 (PSMB5) and β5i (PSMB8). (**B**). RT-PCR analysis of β5i (PSMB8) mRNA expression. HFFs: human foreskin fibroblasts; iPSCs: induced pluripotent stem cells; NSCs: neural stem cells; MN: motor neuron.

**Table 1 cells-12-02800-t001:** List of cells used in the study.

All Human Cells	Cell List	Cell Type	Passage	Assay
Fibroblasts	HFF1	Human forskin fibroblast	p2	Proteomic & Western Blot
(4 samples)	HFF2	Human forskin fibroblast	p3	Proteomic & Western Blot
	SP15	Human skin fibroblast	p4	Proteomic & Western Blot
	CHB	Human skin fibroblast	p4	Proteomic & Western Blot
iPSC	DH1	synthetic mRNA made iPSC	p9	Proteomic & Western Blot
(8 samples)	DH3	synthetic mRNA made iPSC	p9	Proteomic & Western Blot
	DH5	synthetic mRNA made iPSC	p11	Proteomic & Western Blot
	MH1	synthetic mRNA made iPSC	p11	Proteomic & Western Blot
	MH6	synthetic mRNA made iPSC	p12	Proteomic & Western Blot
	MH9	synthetic mRNA made iPSC	p12	Proteomic & Western Blot
	iPSC1	Retroviral made iPSC	p10	Western Blot
	iPSC2	Retroviral made iPSC	p10	Western Blot
NSC	DH1	derived neural stem cells	p4	Proteomic & Western Blot
(3 samples)	DH3	derived neural stem cells	p4	Proteomic & Western Blot
	DH5	derived neural stem cells	p4	Proteomic & Western Blot
Motor Neurons	DH1	Derived Motor Neurons	p3	Western Blot
(3 samples)	DH3	Derived Motor Neurons	p3	Western Blot
	DH5	Derived Motor Neurons	p3	Western Blot

## Data Availability

The full proteomic data are available in an Excel file in the Appendix A.

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
