# Peer review of "Proteomic Analysis of Human iPSC-Derived Neural Stem Cells and Motor Neurons Identifies Proteasome Structural Alterations"

_cells, 2023, doi:10.3390/cells12242800_

Round 1
Reviewer 1 Report
Comments and Suggestions for Authors
Figure 1: Authors can split the figure showing karyotype, data is not visible clearly.
Figure2: Please mention what are the two figures for a single clone representing. Are these before & after differentiation? Can these panel be split up for better presentation? Expression of proteins labelled by red color is hardly visible.
Figure3: Figure is blurry possibly because of low resolution, please replace it with a high quality. Numbers on the figures are hard to read. (B) What are the different colors for action potential (Red, Green, Blue, Black) representing ?
Result:
3.2.1: Even though authors are studying expression of UPS proteins by measuring proteome it is worth trying to measure the levels at transcriptome (mRNA levels). This will explain if there is overall reduced expression or there is high protein turnover upon iPSC formation.
line 250: Authors have used term "strong downregulation" of PSME1, it will be more appropriate to mention how much is the fold change. Sae goes for line 253 where term "significant upregulation" is being used.
Line 320: There is no data supporting the statement (B1i and B2i could not be detected in fibroblasts) hence it should be removed.
Figure6: The blot for Beta 5 should be replaced. Though authors have quantified the ration, blot quality is not suitable for quantification. Also, visually expression of Beta5 seems comparable or maybe higher in HFF lysate compared to MN which contradicts the conclusion.
Discussion:
Misfolded proteins in neuronal disease: This discussion seems to be a bit far fetched based on findings in the manuscript. It should be toned down. The MN model used in this study does not represent ALS MN model, why authors have linked their findings to MHC1 activation in ALS MN?
Comments on the Quality of English LanguageComment: Some sentences in the manuscript are very lengthy, which makes it hard to understand.
Authors have not spent enough time for manuscript proofreading hence there are typos.
Line 244: typo "processed" please replace.
Line 357: typo "that" please replace.
Author Response
Reviewer 1
Comments and Suggestions for Authors
Figure 1: Authors can split the figure showing karyotype, data is not visible clearly.
RESPONSE: Figure 1 line 248: We thank the reviewer for this comment and have added a higher resolution of the karyotype in the figure to address this point and made it bigger, however this the best resolution we were allowed from the software used for karyotyping at the time.
Figure2: Please mention what are the two figures for a single clone representing. Are these before & after differentiation? Can these panel be split up for better presentation? Expression of proteins labelled by red color is hardly visible.
RESPONSE: Figure 2 Line 253-274: We thank the reviewer for this clarification. There are six clones characterized in this figure. We have added a sentence to the figure 2 legend to clarify this point and reorganized the figure. We agree that the Sox2 and Olig2 expression in red is weaker than the Nestin and TUJ1 expression (green) and represents what was observed.
Figure3: Figure is blurry possibly because of low resolution, please replace it with a high quality. Numbers on the figures are hard to read. (B) What are the different colors for action potential (Red, Green, Blue, Black) representing ?
RESPONSE: Figure 3 Line 278: We thank the reviewer for this clarification. We have improved the numbers in the figure and added a more detailed explanation in the figure legend.
Result:
3.2.1: Even though authors are studying expression of UPS proteins by measuring proteome it is worth trying to measure the levels at transcriptome (mRNA levels). This will explain if there is overall reduced expression or there is high protein turnover upon iPSC formation.
RESPONSE: Figure 6 line 377: This a very salient point raised by the reviewer, and we thank the reviewer for the suggestion. We have performed RTPCR for Beta5i mRNA expression and have added this data to figure 5, as suggested by the reviewer. The data shows a significant reduction in B5i gene expression from HFF to iPSC or NSC but an observable increase in MN and 50 day old MN, following the protein expression profile.
line 250: Authors have used term "strong downregulation" of PSME1, it will be more appropriate to mention how much is the fold change. Sae goes for line 253 where term "significant upregulation" is being used.
RESPONSE line 298 and 302: We thank the reviewer, and this has been corrected in the text.
Line 320: There is no data supporting the statement (B1i and B2i could not be detected in fibroblasts) hence it should be removed.
RESPONSE Line 359 and new Supl. Figure 4: We thank the reviewer and have corrected this by placing the Western Blot data for Beta1i and Beta2i with an analysis of mRNA expression of both genes by RT-PCR in a new supplementary Figure 4 and correction in the text at line 360. Although the western blot loading control for Beta2i did not work well, it did work very well for Beta1i and we repeated both blots and in each case the Beta1i and Beta2i protein could not be detected. We suggest that the binding affinity of the antibodies for Beta1i and beta 2i are too weak, or protein expression levels are too low in these cells for these two immunoproteasome subunits.
Figure6: The blot for Beta 5 should be replaced. Though authors have quantified the ration, blot quality is not suitable for quantification. Also, visually expression of Beta5 seems comparable or maybe higher in HFF lysate compared to MN which contradicts the conclusion.
RESPONSE Line 377 and Figure 6: This is an important point raised by the reviewer and have modified the conclusion. We have repeated the Western Blot for Beta 5 twice to try and improve the western blot but were unable to improve the picture. We agree with the reviewer that the quality of the Western Blot is not “textbook”, but is a result that we face every day in the laboratory and is semi-quantifiable (not quantifiable). To strengthen this aspect of the work we have added RT-PCR mRNA expression analysis of beta 5i subunit (PSMB8) to Figure 6 that supports the Western Blot data that beta5i subunit expression decreases from HFF to iPSC/NSC but then increases from iPSC/NSC to MN or 50 day differentiated MN (Figure 6). Further RT-PCR data presented in Supl figure 4 for Beta1i and beta2i mRNA expression demonstrates the same trend as beta 5i. From Figure 6, the ratio of CP to IP changes in HFF to MN (the CP is higher expressed) HFF to MN is comparable, however our conclusion is divided into 2 parts: A decrease in immunoproteasome subunit expression beta 1i, 2i and 5i from HFF to iPSC/NSC and then an increase from iPSC/NSC to MN with an overall higher level of expression of CP in MN.
Discussion:
Misfolded proteins in neuronal disease: This discussion seems to be a bit far fetched based on findings in the manuscript. It should be toned down. The MN model used in this study does not represent ALS MN model, why authors have linked their findings to MHC1 activation in ALS MN?
RESPONSE: We thank the reviewer for this point, and we have re-written this part of the discussion to keep closer to the data and design of the project. Also, we have deleted a part of the first sentence on line 373 of the discussion as it was a typographical error repeating the same point.
Comments on the Quality of English Language
Comment: Some sentences in the manuscript are very lengthy, which makes it hard to understand.
Authors have not spent enough time for manuscript proofreading hence there are typos.
Line 244: typo "processed" please replace.
Line 357: typo "that" please replace.
RESPONSE: We thank the reviewer for pointing these typos out to use, they have been corrected and we have gone through the manuscript to reduce sentence length and any other typos that we can find.
Reviewer 2 Report
Comments and Suggestions for Authors
Neural stem cells (NSCs) provide a means to study normal neural development, neurodegeneration, and neurological disease. The discovery of the method to make human induced pluripotent stem cells (iPSC) created the opportunity to model human disease and development at a new level. Proteomics studies of NSCs have the potential to delineate molecules and pathways critical for NSC biology and how NSCs can participate in neural repair. In this study, the author used iPSC differentiated to neural stem cells and then to motor neurons as a developmental model, and the iPSC clones were successfully differentiated to motor neurons (MN). The author then performed a proteomic analysis of skin fibroblasts, iPSC, differentiated neural stem cells (NSC), and motor neurons (MN). Hundreds to thousands of proteins had been found to be differentially regulated in iPSC and NSC compared to skin fibroblasts, involving in various biological processes, including protein degradation by the proteasome, which was validated via immunoblotting analysis. However, there are still many issues that need to be addressed before acceptance.
Comments:
1. The detailed conditions for mass samples were missing, such as the working concentration of trypsin used for protein digestion, digestion time, desalting et al.
2. The detailed setting parameters for LC-MS/MS analysis and proteomic data analysis were missing.
3. All the figures lacked detailed descriptions of corresponding data.
4. In Figure 1, there was no mark for 1a-1d. In Figure 1c (supposed to be based on the text), the error bar should be provided.
5. In Figure 1c, the gene names should be consistent with the ones mentioned in the text.
6. In Figure 2, the author didn’t mention the cells used for analysis, HFF, iPSC, NSC, or MN. Zoomed-in picture should be added to highlight the expression of marker proteins. Separated channels should be provided for each result.
7. The resolution of Figure 3 was too low. There was no information provided for what the lines with different colors stood for.
8. After proteomic raw data analysis, the author should perform KEGG and GO analysis for all experimental groups.
9. The title and text were not consistent in section 3.2.2. The author should claim whether it was a comparison between NSC and HFF or iPSC and NSC.
10. In Figure 4, the protein dots with -log10(P value) less than 1 should also be presented. And the representative proteins especially those mentioned in the text should be labeled in the volcano plots.
11. It would be more reliable to set the cutoff for proteome difference at 1.5 instead of 1.
12. The order of Figure 5 and Figure 6 should be changed based on the text.
Comments on the Quality of English LanguageThe gene names should be consistent in the paper.
Author Response
Reviewer 2
Neural stem cells (NSCs) provide a means to study normal neural development, neurodegeneration, and neurological disease. The discovery of the method to make human induced pluripotent stem cells (iPSC) created the opportunity to model human disease and development at a new level. Proteomics studies of NSCs have the potential to delineate molecules and pathways critical for NSC biology and how NSCs can participate in neural repair. In this study, the author used iPSC differentiated to neural stem cells and then to motor neurons as a developmental model, and the iPSC clones were successfully differentiated to motor neurons (MN). The author then performed a proteomic analysis of skin fibroblasts, iPSC, differentiated neural stem cells (NSC), and motor neurons (MN). Hundreds to thousands of proteins had been found to be differentially regulated in iPSC and NSC compared to skin fibroblasts, involving in various biological processes, including protein degradation by the proteasome, which was validated via immunoblotting analysis. However, there are still many issues that need to be addressed before acceptance.
Comments:
- The detailed conditions for mass samples were missing, such as the working concentration of trypsin used for protein digestion, digestion time, desalting et al.
RESPONSE line115 to 152: We thank the reviewer, and this has been corrected in the text.
- The detailed setting parameters for LC-MS/MS analysis and proteomic data analysis were missing.
RESPONSE line 115 to 152: We thank the reviewer, and this has been corrected in the text.
- All the figures lacked detailed descriptions of corresponding data.
RESPONSE: We have modified the figures and legends throughout the manuscript. We have also added text to the discussion.
- In Figure 1, there was no mark for 1a-1d. In Figure 1c (supposed to be based on the text), the error bar should be provided.
RESPONSE line 248 to 252: We have reorganized Figure 1 and added labels A to D.
- In Figure 1c, the gene names should be consistent with the ones mentioned in the text.
RESPONSE line 240: We have gone through the text and ensured that the gene names are consistent.
- In Figure 2, the author didn’t mention the cells used for analysis, HFF, iPSC, NSC, or MN. Zoomed-in picture should be added to highlight the expression of marker proteins. Separated channels should be provided for each result.
RESPONSE line 271. We have included in the figure 2 legend that six motor neuron (MN) clones were characterized for neuron stem cell and and MN markers by IF at high magnification.
- The resolution of Figure 3 was too low. There was no information provided for what the lines with different colors stood for.
RESPONSE line 278: We have improved the resolution of the labeling to make clearer in Figure 3, and added text to the figure legend to explain fully the figure.
- After proteomic raw data analysis, the author should perform KEGG and GO analysis for all experimental groups.
RESPONSE: We thank the reviewer for this comment and agree that further analysis could be performed. Here, we have performed volcano plots of the data processed using a human spectral meta-library in Proteomics Discoverer (Version 2.1, Thermo Scientific). Indeed, many proteins were identified and are presented in the Excel in the supplementary section freely open for further analyses.
- The title and text were not consistent in section 3.2.2. The author should claim whether it was a comparison between NSC and HFF or iPSC and NSC.
RESPONSE line 316: We have corrected the title in 3.2.2.
- In Figure 4, the protein dots with -log10(P value) less than 1 should also be presented. And the representative proteins especially those mentioned in the text should be labeled in the volcano plots.
RESPONSE: This is a good point raised by the reviewer, the analysis in Figure 4 was performed by the proteomic platform at the Helmholt Institute and they set it to their default settings. This is meant as a visual representation of the proteomics and in Figure 6 we have continued the analysis and labeled the proteasome proteins in that figure.
- It would be more reliable to set the cutoff for proteome difference at 1.5 instead of 1.
RESPONSE: We thank the reviewer for this suggestion and agree a higher cutoff for proteome difference would result in more rigid set of proteins. However, the approach we wanted to take is to widen the catchment area to include small differences that are then validated by western blot analysis. The raw proteomic data is available in the supplementary figure section for further abalysis by the public if needed.
- The order of Figure 5 and Figure 6 should be changed based on the text.
RESPONSE: We thank the reviewer for this comment and feel that figure 5 is a more general picture of the proteomic screen while figure 6 follows on with more details of the proteasome sub units expressed.
Comments on the Quality of English Language
The gene names should be consistent in the paper.
RESPONSE line 240: We thank the reviewer and have gone through the text again to ensure the gene names are the same.
Reviewer 3 Report
Comments and Suggestions for Authors
The authors of this study have presented an intriguing investigation into the role of the proteasome in motor neuron development, focusing on proteomics analysis and the identification of changes in proteasome subunit expression during differentiation from induced pluripotent stem cells (iPSCs) to motor neurons. The research sheds light on potential implications for neurodevelopmental diseases. However, several key points in the methods and analysis warrant further discussion and clarification.
First and foremost, the authors have employed a quantitative proteomics approach using normal trypsin and Lys-C digestion to elucidate the protein profile under different experimental conditions. While this methodological choice is understandable, the study could significantly benefit from a more detailed discussion of the rationale behind this selection. Given the critical role of the proteasome in protein degradation and its implications for the cellular proteome, it is pertinent to address whether the chosen proteomic analysis specifically targeted proteasome substrates or the changes in proteasome-generated peptides. Clarifying this aspect would enhance the comprehensibility of the study.
One notable concern that arises from this work is the apparent absence of peptidomics analysis, which would be invaluable for characterizing the immune peptides generated by the proteasome in different cellular states. These immune peptides play a crucial role in immune responses and could be integral to the understanding of the proteasome's involvement in motor neuron development. An explanation for the omission of peptidomics data is warranted, and its inclusion would provide a more comprehensive view of the proteasome's involvement.
Furthermore, the article briefly mentions MHC peptide enrichment without providing substantial details on how this enrichment was achieved. This is a crucial aspect, as MHC peptide analysis is pivotal for understanding the immunoproteasome's role and its implications. A more in-depth description of the methods used for MHC peptide enrichment is essential for the reader's understanding and for the reproducibility of the study.
In summary, this article presents valuable insights into the proteasome's role in motor neuron development and its potential implications for neurodevelopmental diseases. However, it is essential to address the points raised regarding the lack of peptidomics analysis and the limited details on MHC peptide enrichment, as these aspects significantly affect the comprehensiveness of the study. Providing a more detailed methodological account and rationale for method choices would enhance the overall quality and impact of the research.
I suggest that the authors conduct either a peptidomics analysis of the secretome to evaluate the peptides generated without in vitro digestion or a MHC peptide enrichment to assess the peptide profile associated with MHC complexes. These additional analyses can provide valuable insights into the role of the proteasome and its generated peptides in motor neuron development and their potential implications for neurodevelopmental diseases.
- Peptidomics Analysis of Secretome: Performing a peptidomics analysis of the secretome would allow the authors to identify and characterize the peptides naturally generated by the proteasome in the cellular context. This analysis can reveal the specific peptide sequences produced by the proteasome during motor neuron development. It is a valuable approach for understanding the immunopeptidome and its potential regulatory role in neuronal plasticity and synapse formation.
- MHC Peptide Enrichment: Another important approach is to perform MHC peptide enrichment. This method allows for the selective isolation and characterization of peptides associated with MHC complexes. Analyzing the peptide profile within MHC complexes can provide insights into the immunopeptidome relevant to motor neuron development. This approach is particularly relevant given the authors' mention of immune-like MHC receptors in the central nervous system.
Including either of these additional analyses would strengthen the study's conclusions and broaden its scope, facilitating a more comprehensive understanding of the proteasome's role in motor neuron development. The data obtained from these analyses can shed light on the nature of peptides generated during the process and their potential impact on neurodevelopmental diseases. Additionally, it would make the study more appealing to the readers and researchers interested in neurobiology and immunology.
I recommend that the authors discuss the feasibility of conducting these additional analyses in their response to reviewers and potentially include them in an expanded version of their research to enrich the study's findings and scientific impact.
Author Response
Reviewer3
(x) English language fine. No issues detected
The authors of this study have presented an intriguing investigation into the role of the proteasome in motor neuron development, focusing on proteomics analysis and the identification of changes in proteasome subunit expression during differentiation from induced pluripotent stem cells (iPSCs) to motor neurons. The research sheds light on potential implications for neurodevelopmental diseases. However, several key points in the methods and analysis warrant further discussion and clarification.
First and foremost, the authors have employed a quantitative proteomics approach using normal trypsin and Lys-C digestion to elucidate the protein profile under different experimental conditions. While this methodological choice is understandable, the study could significantly benefit from a more detailed discussion of the rationale behind this selection. Given the critical role of the proteasome in protein degradation and its implications for the cellular proteome, it is pertinent to address whether the chosen proteomic analysis specifically targeted proteasome substrates or the changes in proteasome-generated peptides. Clarifying this aspect would enhance the comprehensibility of the study.
RESPONSE We thank the reviewer for the insight and comment. We have contacted the Proteomics service in Germany we used to answer this question:
“Lys-C and trypsin as proteases are widely used in proteomics LC-MSMS approaches, due to their high cleavage efficiencies, the reasonable sizes of generated peptides and the advantage of resulting charge states of the peptides for mass spectrometric measurements. We use them in almost all our projects.”
Regarding if we should “address whether the chosen proteomic analysis specifically targeted proteasome substrates or the changes in proteasome-generated peptides.”:
“we feel that it definitely didn’t – assuming the proteasome was active and introduced cleavage sites in proteins to be degraded – and then we added trypsin on the whole protein mixture, the result would be so-called semi-tryptic peptides (meaning on one end of the peptide a trypsin-induced cut and on the other side a non-tryptic proteasomal cut). These kinds of peptides are not included in the spectral library which we used in the search. So even if these peptides would be generated in a reasonable amount (which I doubt), we would not we able to identify them with the given approach.”
One notable concern that arises from this work is the apparent absence of peptidomics analysis, which would be invaluable for characterizing the immune peptides generated by the proteasome in different cellular states. These immune peptides play a crucial role in immune responses and could be integral to the understanding of the proteasome's involvement in motor neuron development. An explanation for the omission of peptidomics data is warranted, and its inclusion would provide a more comprehensive view of the proteasome's involvement.
RESPONSE: I agree with the reviewer that the identification of the immunopeptidome of the MHC-I will provide very important information on the motor neuron development and the putative targets of the T cell immune response in neurological diseases. However, the complexity of the MHC-I immunopeptidome, the high degree of polymorphism of MHC genes and the low MHC-I expression in motor neurons make this analysis complex. In addition, the proteomics facilities are expensive. Altogether, to perform a correct analysis of the MHC-I immunopeptidomes, different biological replicates from different donors together with larger number of cells, require proteomics analysis performed and additional funding would be required. Thus, future analysis of the MHC-I peptide repertoires must be performed, but this is beyond the scope of this current work. We consider that the data presented in this work should guarantee obtaining additional funding to make the analysis proposed by the referee.
Furthermore, the article briefly mentions MHC peptide enrichment without providing substantial details on how this enrichment was achieved. This is a crucial aspect, as MHC peptide analysis is pivotal for understanding the immunoproteasome's role and its implications. A more in-depth description of the methods used for MHC peptide enrichment is essential for the reader's understanding and for the reproducibility of the study.
RESPONSE: We agree that MHC peptide enrichment is a crucial step; however, no peptide enrichment was made in this work.
In summary, this article presents valuable insights into the proteasome's role in motor neuron development and its potential implications for neurodevelopmental diseases. However, it is essential to address the points raised regarding the lack of peptidomics analysis and the limited details on MHC peptide enrichment, as these aspects significantly affect the comprehensiveness of the study. Providing a more detailed methodological account and rationale for method choices would enhance the overall quality and impact of the research.
RESPONSE: See answer in 1 and 2.
I suggest that the authors conduct either a peptidomics analysis of the secretome to evaluate the peptides generated without in vitro digestion or a MHC peptide enrichment to assess the peptide profile associated with MHC complexes. These additional analyses can provide valuable insights into the role of the proteasome and its generated peptides in motor neuron development and their potential implications for neurodevelopmental diseases.
- Peptidomics Analysis of Secretome: Performing a peptidomics analysis of the secretome would allow the authors to identify and characterize the peptides naturally generated by the proteasome in the cellular context. This analysis can reveal the specific peptide sequences produced by the proteasome during motor neuron development. It is a valuable approach for understanding the immunopeptidome and its potential regulatory role in neuronal plasticity and synapse formation.
RESPONSE: I agree that the analysis of the secretome (and exosomes) is of great interest to study the generation and processing of the intracellular proteome. Nevertheless, the secretome is, probably, produced in the vesicular pathway (although it is possible that some cytosolic proteins captured by autophagy can be present). Thus, in our opinion, it is not expected to secretome contains many peptides generated by the proteasome.
- MHC Peptide Enrichment: Another important approach is to perform MHC peptide enrichment. This method allows for the selective isolation and characterization of peptides associated with MHC complexes. Analyzing the peptide profile within MHC complexes can provide insights into the immunopeptidome relevant to motor neuron development. This approach is particularly relevant given the authors' mention of immune-like MHC receptors in the central nervous system.
RESPONSE: See answer in 1.
Including either of these additional analyses would strengthen the study's conclusions and broaden its scope, facilitating a more comprehensive understanding of the proteasome's role in motor neuron development. The data obtained from these analyses can shed light on the nature of peptides generated during the process and their potential impact on neurodevelopmental diseases. Additionally, it would make the study more appealing to the readers and researchers interested in neurobiology and immunology.
I recommend that the authors discuss the feasibility of conducting these additional analyses in their response to reviewers and potentially include them in an expanded version of their research to enrich the study's findings and scientific impact.
RESPONSE: We agree with the referee. These additional analyses are of outstanding interest to define the relevance of the variation in the proteasome complexes. Nevertheless, as said above, these analyses require high number of cells, variation in the samples and are time consuming, beyond the scope of this paper. We are planning and writing a grant to perform these experiments in the near future and agree with the reviewer are very interesting and important.
Round 2
Reviewer 1 Report
Comments and Suggestions for Authors
Authors have taken care of major comments, Manuscript is improved.
Comments on the Quality of English LanguageMinor editing.
Author Response
2nd round RESPONSE to the comments
Reviewer 1
- Authors have taken care of major comments, Manuscript is improved.
RESPONSE: On behalf of the authors, we thank the reviewer for the constructive comments that has resulted in a much-improved manuscript.
- Comments on the Quality of English Language: Minor editing.
RESPONSE: We have read through the manuscript and checked the English again and made several corrections.
Reviewer 2 Report
Comments and Suggestions for Authors
1. The presence of initialisms, like MN, HFF, and NSC et. al., full names should be in the first instance and followed immediately by the abbreviated version in brackets.
2. Figures 1, 2, and 6 are unreadable, and the resolution of Figure 3 is still too low.
3. The figure legend "Protocol to generate motor neurons from human iPSC and characterisation" is not precise.
4. It is better to label some representative proteins in Figure 4.
Comments on the Quality of English Language
Minor editing of English language required
Author Response
2nd round RESPONSE to the comments
Reviewer 2
- The presence of initialisms, like MN, HFF, and NSC et. al., full names should be in the first instance and followed immediately by the abbreviated version in brackets.
RESPONSE: We thank the reviewer and have added to all the figure legends the full name for the initials used in the figures, leading to a clearer manuscript, thank you.
- Figures 1, 2, and 6 are unreadable, and the resolution of Figure 3 is still too low.
RESPONSE: We apologize for this oversight and have increased the font to make it more readable. For Figure 3, the person who generated that data is no longer available to provide a better resolution, however we have improved the numbers and the text in the figure so it is clearer. This has led to improved figures and on behalf of the authors we thank the reviewer for this comment.
- The figure legend "Protocol to generate motor neurons from human iPSC and characterisation" is not precise.
RESPONSE line: We thank the reviewer for this comment and have improved the legend figure and text to clarify better the protocol.
- It is better to label some representative proteins in Figure 4.
RESPONSE: The proteomics service we used in Germany is unable to provide further data analysis for Figure 4, as it is out of scope of the budget provided. Proteins of interest in this studied are labeled in the Figure 5 volcano plot. The raw global proteomic data is also added as an excel file for free public access.
- Comments on the Quality of English Language: Minor editing of English language required
RESPONSE: We have read through the manuscript and checked the English again and made several corrections.